



# Multimethod determination of the below-cloud wet scavenging coefficients of aerosols in Beijing, China

Danhui Xu[1,2], Baozhu Ge[1,*], Xueshun Chen[1], Yele Sun[1], Nianliang Cheng[3], Mei Li[4,5], Xiaole Pan[1], Zhiqiang Ma[6], Yuepeng, Pan[1], Zifa Wang[1]

5  [1]State Key Laboratory of Atmospheric Boundary Layer Physics and Atmospheric Chemistry, Institute of Atmospheric Physics, Chinese Academy of Sciences, Beijing 100029, China
[2]University of Chinese Academy of Sciences, Beijing 100049, China
[3]Beijing Municipal Environmental Monitoring Center, Beijing 100048, China
[4]Institute of Mass Spectrometer and Atmospheric Environment, Jinan University, Guangzhou 510632, China
10  [5]Guangdong Provincial Engineering Research Center for on-line source apportionment system of air pollution, Guangzhou 510632, China
[6]Beijing Shangdianzi Regional Atmosphere Watch Station, Beijing 100089, China

*Correspondence*: gebz@mail.iap.ac.cn



**Abstract.** Wet scavenging is one of the most efficient processes that remove aerosols from the atmosphere. This process is not well constrained in chemical transport models (CTMs) due to a paucity of localized parameterization regarding below-cloud wet scavenging coefficient (BWSC). Here we conducted field measurements of the BWSC during the Atmospheric Pollution and Human Health-Beijing (APHH-Beijing) campaign of 2016. Notably, the observed BWSC values based on the updated aerosol mass balance agree well with another estimation technique by the updated aerosol mass balance, and they fall in a range of $10^{-5}$ $s^{-1}$. The measurement in this winter campaign, combined with that in summer of 2014, supported an exponential power distribution of BWSCs with rainfall intensity. The observed parameters were also compared with both the theoretical calculations and modeling results. We found that the theoretical estimations can effectively characterize the observed BWSCs of aerosols with size smaller than 0.2 μm and larger than 2.5 μm. However, the theoretical estimations were one magnitude lower than observed BWSCs within 0.2-2.5 μm, a domain size range of urban aerosols. Such an underestimation of BWSC through theoretical method has been confirmed not only in APHH-Beijing campaign but also in all the rainfall events in summer of 2014. Since the model calculations usually originated from the theoretical estimations with simplified scheme, the significant lower BWSC would well explain the underprediction of wet depositions in polluted regions as reported by the Model Inter-Comparison Study for Asia (MICS-Asia) and the global assessment of the Task Force on Hemispheric Transport of Atmospheric Pollutants (TF-HTAP). The findings highlighted that the wet deposition module in the CTMs requires improvement based on field measurement estimation to construct a more reasonable simulation scheme for BWSC, especially in polluted regions.

# 1 Introduction

Wet deposition is one of the dominant aerosol sinks on both global and regional scales (Min et al., 2005;Textor et al., 2006), and can be divided into in-cloud (particles are activated as cloud condensation nuclei and absorbed by cloud water) and below-cloud scavenging (aerosols and gas are captured by raindrops or snow particles after the hydrometeors leave the clouds) (Zhao et al., 2015). Previously, below-cloud scavenging is thought to be less important than in-cloud process and always simplified or even ignored in most global and regional chemical transport models (CTMs) (Tang et al., 2006;Bae et al., 2010;Barth et al., 2000;ENVIRON.Inc, 2005;Stier et al., 2005). This may be true in most clean atmosphere, e.g., developed country where air pollutants in the boundary layer were not sufficient. This may be not the case in polluted regions. Recently, regional model in MICS-Asia (Model Inter-Comparison Study for Asia) has found the obvious underestimation of wet deposition in East Asia (Wang et al., 2008). For global model assessment by Hemispheric Transport of Atmospheric Pollutants (TF-HTAP), wet depositions are also underpredicted in region of North America, Europe and Asia where measured the high level of volume weighted averaged (VWA) nitrogen (N) concentrations in rainfall as > 1.25 mg N $L^{-1}$ (Vet et al., 2014). Besides the uncertainties in emission inventory and chemical mechanism, the below-cloud scavenging process may also contribute to certain effects on the wet deposition simulation (Wang et al., 2008). Actually, below-cloud scavenging cannot be negligible in



CTMs, which contributed to more than 53% of the total wet deposition in some polluted areas such as India (Chatterjee et al., 2010) and North China (Ge et al., 2016;Xu et al., 2017).

Extensive efforts have been focused on the study of wet scavenging, and many researchers have noted that precipitation, even light rain, can remove 50-80% of the number or mass concentration of below-cloud aerosols (Andronache, 2004b;Zhang et al., 2004). The below-cloud wet scavenging coefficient (hereafter, BWSC), denoted $K$ or $K(d_p)$ for size-resolved values, is a parameter that describes scavenging ability characteristics fairly well. The main factors affecting the BWSC include raindrop number size distribution, collection efficiency and raindrop terminal velocity, remain un-known and hence make the large uncertainties of BWSC (Wang et al., 2010). Seinfeld and Pandis (1998) proposed that collection efficiency (Brownian

diffusion, directional interception, inertial impaction, thermophoresis and diffusion electrophoresis) is critical in the below-cloud scavenging process. Coarse particles (aerosol particle sizes $d_p$ ranging from 2-20 μm) are easily scavenged by inertial impaction. Especially coarse particles ($d_p > 20$ μm) are also easily scavenged through the effect of gravity. Fine particles ($d_p < 0.2$ μm) can be removed by Brownian diffusion. However, accumulation mode aerosols (0.2 μm $< d_p <$ 2 μm) are neither efficiently scavenged by Brownian diffusion nor by directional interception or inertial impaction, and this particle size range

is called the "Greenfield gap" (Slinn, 1984). Recently, Bae et al. (2010) added phoretic and electric charging effects to the collection efficiency assessment and found that the BWSCs increase by up to 20 times in the 0.2-3 μm particle size range. Wang et al. (2014c) also improved the understanding of the electrical effects of the collision efficiency, which is also assumed to be a major source of uncertainty but is always ignored in theoretical estimations. It also improved the BWSC estimation by an order of magnitude. The raindrop number size distribution and raindrop terminal velocity are both represented by empirical

mathematical functions, and these factors are non-negligible. In order to minimize the computational burden, the calculation of BWSCs in most global and regional-scale models are expressed as the product of rain intensity multiplied by the collection efficiency, where the later is simplified as a constant or calculated based on the work of Slinn (Bae et al., 2010;Slinn, 1984). This simplification may undoubtedly bring into large uncertainties and make the simulated wet deposition within a factor of two ranges of the observations, which is significant larger compared with the 30% bias of the prediction of air pollutants

evaluation (Vet et al., 2014;Zhu et al., 2018).

    Over the past few decades, a lot of wet scavenging coefficient (WSC) field measurements have been the focus of a large number of studies (Andronache, 2004b;Jylhä, 1991;Laakso et al., 2003;Okita et al., 1996;Wang et al., 2014c;Xu et al., 2017). In their field measurements, Okita et al. (1996) used the precipitation intensity, cloud-base height and the ratio between the sulfate concentration in aerosols of air mass and in rainwater to estimate the WSC, and this method is widely applied in most

75 field measurements at present (Andronache, 2004b;Yamagata et al., 2009). However, this method cannot distinguish the below cloud part from the whole wet scavenging process, which is important to the parameterization scheme in CTMs. Xu et al. (2017) adopted sequential sampling and estimated the BWSCs of various soluble inorganic ions using the washout fraction concentration. In addition, the BWSCs display a strong dependency on the aerosol particle size distribution. Laakso et al. (2003) indicated that the BWSCs could be calculated by the aerosol particle number concentrations for various size ranges





both before and after rain. This size-resolved method was also applied in Lanzhou (Zhao et al., 2015), Huang Mountain (Wang et al., 2014c), China, Southern Finland (Andronache et al., 2006) and India (Chate et al., 2003). In general, both methods are widespread for the estimation of WSCs/BWSCs but few were focused on the differences among these methods.

In this study, we compare the WSCs/BWSCs estimated from original and updated observational methods with the theoretical and model calculations under the same conditions to perform a multimethod evaluation to describe its characteristics. First,

we introduce the basic circumstances of the data collected with multiple observation instruments. Then, we present the various methods, compare the results and discuss the discrepancies among the different results. Finally, we evaluate the effect of below-cloud scavenging on aerosol concentrations and wet depositions based on multimethod techniques.

## 2 Methods and Data

### 2.1 Sampling site and measurement data

In North China, precipitations were mainly concentrated in summer but rarely in autumn and winter (Xu et al., 2008, Gao et al., 2015). However, the air pollution events were usually occurred in autumn and winter in North China Plain (NCP). Here we select a typical rainfall event in winter of Beijing (a typical air polluted city in NCP) to study the characteristic of BWSC and its implications to aerosol scavenging. The sampling site is situated on top of the two-floor building of the Institute of Atmospheric Physics (IAP, 39°58′28″N, 116°22′1″E), located between the north $3^{rd}$ and $4^{th}$ Ring Road of Beijing. The site is

a typical urban site and is 1 km away from the main road to the north and east, near residential buildings to the south and a park to the west, and the pollution at this site is mainly from traffic and domestic sources (Sun et al., 2015). The selected rainfall case lasts from 6:56 AM on November $20^{th}$ to 1:18 AM on November $21^{st}$, 2016, which is during the wintertime Atmospheric Pollution and Human Health-Beijing (APHH-Beijing) campaign of 2016 (Shi et al., 2018). Thus, comprehensive measurements of air pollutants and simulations of pollution mechanisms are available for our use to investigate the wet

scavenging process. Fig. s1 shows the radar base reflectivity with echo coverage over the urban area of Beijing from 11:54 AM on November $20^{th}$, and gradually moving from northwest to southeast. The total recorded amount of rainfall is 5.2 mm, and the rainfall is more concentrated on the evening of November $20^{th}$ (beginning at 16:29 PM on November $20^{th}$) during this event. Notably, the rainfall is nonuniform across Beijing (Song et al., 2015). For example, the recorded rainfall in the southern suburban area of Beijing is approximately 7 mm according to the Beijing Meteorological Administration, and this rainfall end

at approximately 9:00 AM on November $21^{st}$. In this study, the precipitation chemistry and aerosol components sampling processes occur at the exact same time at the APHH measurement site.

An automatic wet-only sequential rainfall sampler is deployed to obtain rainfall samples with 1 mm increments in one precipitation event. Four anions ($SO_4^{2-}$, $NO_3^-$, $Cl^-$ and $F^-$) and five cations ($NH_4^+$, $Na^+$, $K^+$, $Ca^{2+}$ and $Mg^{2+}$) in these samples are measured by ion chromatography (IC, Dionex 600, USA). The VWA concentrations of the major soluble inorganic ions, i.e.

$NO_3^-$, $SO_4^{2-}$ and $NH_4^+$ (hereafter, SNA) in this rainfall are 35.8, 48.7 and 17.5 mg. $L^{-1}$, respectively, and much higher than the VWA concentrations in the winter of 2016 (8.3, 9.5 and 4.1 mg. $L^{-1}$, respectively) and in previous studies in Beijing (Pan et



al., 2013, 2012;Xu et al., 2017). An ambient ion monitor-ion chromatograph (AIM-IC) developed by URG Corp., Chapel Hill, NC and Dionex Inc., Sunnyvale, CA, is used to measure the PM$_{2.5}$ composition. The time resolution is 60 min. A detailed description of the measured concentration in the rainfall and aerosols can be found in Xu et al. (2017).

Thirty meters away from the sampling site, a scanning mobility particle sizer (SMPS) is deployed to observe the particle number size distribution with a 5-min time resolution. The SMPS is used to measure particle number concentration from 14 to 685 nm. A detailed description of the SMPS and methods can be found in Du et al. (2017).

A single-particle aerosol mass spectrometer (SPAMS), which can accurately characterize aerosol particles containing various chemical compositions with diameters ranging from 0.2 to 2.5 μm, is also deployed during the measuring time. More
detailed fundamentals of the SPAMS and description can be found in Li et al. (2011), Lin et al. (2017) and Cheng et al. (2018). Size-resolved airborne NO$_3^-$, SO$_4^{2-}$ and NH$_4^+$ are the main focuses in this study, and the time resolution is 1 hour. In the meantime, a polarization optical particle counter (POPC) is also deployed to obtain coarse particle (0.4-10.35 μm) size distribution, and time resolution is 5-min. Detailed description and settings can be found in Tian et al. (2018).

## 2.2 Methods

### 2.2.1 Theoretical basis

Seinfeld and Pandis (1998) proposed the following basic equation of variation of the particle number concentration $N(d_p)$:

$$\frac{dN(d_p)}{dt} = -K(d_p)N(d_p) \tag{1}$$

This equation considers that there is no chemical reaction or emission, and wet scavenging is an exponential process. $d_p$ is the diameter of the aerosol particle, and $K(d_p)$ is the size-resolved BWSC obtained by the following equation:

$$K(d_p) = \int_0^\infty \frac{\pi}{4} D_p^2 U_t(D_p) E(D_p, d_p) N(D_p) dD_p \tag{2}$$

where $D_p$ is the raindrop diameter. $U_t(D_p)$ and $N(D_p)$ are the falling terminal velocity and concentration of raindrops, respectively. There are two approaches for describing $U_t(D_p)$: an empirical formula and a physically based formula. Many expressions have been employed for various raindrop diameter ranges. In addition, there are still no available mathematical functions that can accurately characterize the natural raindrop size spectra, and exponential, gamma and lognormal
distributions are still used to represent $N(D_p)$ (Wang et al., 2010). Marshall and Palmer (1948) proposed the M-P distribution of raindrop size distribution, which is mostly applied to calculations of BWSCs. $E(D_p, d_p)$ is the collision efficiency of raindrops and aerosol particles, which, in most studies, mainly involves Brownian diffusion, interception and inertial impaction due to dimensional analysis without accounting for thermophoresis, diffusiophoresis and electric charges (Slinn, 1984;Wang et al., 2010). An extensive number of studies have realized that using only the three main mechanisms results in





underestimation of the collision efficiency, and the contributions from the other mechanisms were added in these studies (Andronache, 2004c;Andronache et al., 2006;Bae et al., 2010). Assuming that a certain size aerosol particle can be captured by raindrops of any size, $K(d_p)$ can be calculated theoretically when the falling terminal velocity, raindrop size distribution and collision efficiency are given. In Wang et al. (2014c)'s study, they added thermophoresis, diffusiophoresis and electric charges to the quantitative calculation, and we considered this updated to be the theory's result.

**2.2.2 Observational method**

In addition to the theoretical calculation, field observations are also critical for estimating BWSCs. One approach is based on the change in the number concentration of aerosols (called M1 in this study). When rainfall occurs from $t_0$ to $t_1$, Eq (1) can be integrated as follows:

$$K(d_p) = \frac{1}{t_1 - t_0} \ln[\frac{N_0(d_p)}{N_1(d_p)}] \tag{3}$$

where $N_0(d_p)$ and $N_1(d_p)$ are the measured aerosol particle number concentrations before the rain occurs ($t_0$) and after the rain ends ($t_1$), respectively (Laakso et al., 2003).

In addition, Andronache (2004b) proposed that the WSC can be estimated by the bulk model based on the aerosol mass balance within a certain bulk, which assumes that there is a box with a horizontal area $A$ and vertical height $h$ above the observation site. The aerosol flux $F$ on the surface per unit time and area is defined as the following equation:

$$F = K \times M \tag{4}$$

where $K$ is the WSC and $M$ is the mass of the aerosols in the given box. $M$ can be described as follows:

$$M = C_a \times A \times h \tag{5}$$

where $C_a$ is the average aerosol concentration in the box.

In addition, $F$ can also be characterized by the following expression:

$$F = C_p \times P \times A \tag{6}$$

where $C_p$ is the aerosol concentration in the precipitation collected at the measurement site, $P$ is the precipitation intensity, and $A$ is the horizontal area for the assumed box. And the wet deposition $D_{ep}$ in a certain time $\Delta t$ can be expressed as:

$$D_{ep} = C_p \times P \times \Delta t = K \times C_a \times h \times \Delta t \tag{7}$$

And $K$ becomes the following expression:

$$K = \frac{C_p}{C_a} \times \frac{P}{h} \tag{8}$$


where $C_p$ and $C_a$ are the paired aerosol concentrations in the precipitation and aerosol, respectively, during rainfall (Okita et al., 1996). In addtion, Andronache (2004b) pointed out that the aerosol concentration in the vertical profile should be considered and updated Eq (8) as follows:

$$K = \frac{C_p}{C_a(0) \times f} \times \frac{P}{h} \tag{9}$$

where $C_a(0)$ is the aerosol concentration at the surface, $h$ is the cloud-base height during rainfall, and

$f = \sum_{z=0}^{z=h} \frac{C_a(z)}{C_a(0)} \times h'(z) / \sum_{z=0}^{z=h} h'(z)$ is the vertical distribution factor of aerosols. Among these variables, $C_a(z)$ are the aerosol

concentrations at the z-level height, respectively, and $h'(z)$ is the depth of the layers in the vertical direction. This approach is called M2.

   Moreover, most studies have mentioned that the prevailing wind in Beijing can efficiently reduce the aerosol concentrations
(Chan and Yao, 2008;Gonzalez and Aristizabal, 2012). In previous studies by Xu et al. (2017), the north and northwest winds have been recognized as the clear streams to scavenge aerosols in situ, and the effects of clean wind is also considered in this study. In addition, with the help of the 1 mm increments sequential rainfall sampling, Xu et al. (2017) has found that the later increments maintained at a stable, low level, which can be separated into rainout process only. Similar with Eq (9), an updated below-cloud estimated method using $C_{p,\text{below}}$ has been developed as Eq (10) and called as M2':

$$K = \frac{C_{p,below}}{C_a'(0) \times f} \times \frac{P}{h} \tag{10}$$

where $C_{p,\text{below}}$ is the washout concentration that  have been eliminated the rainwater concentrations in each increment and $C_a'(0)$ is the aerosol concentration at the surface that considered the eliminated effects of north and north-west wind.

### 2.2.3 Modeling calculation

In this study, the Nested Air Quality Prediction Modeling System (NAQPMS) was adopted to calculate the aerosol scavenging
coefficient. The RADM-2 mechanism is used for aqueous-phase chemistry, and the subgrid-scale vertical redistribution, dissolution, dissociation and wet deposition are included (Chang et al., 1987). The wet scavenging module used in this study mainly includes below-cloud scavenging process, and the wet scavenging process is briefly described as follows:

$$K = \frac{10.8 \times E \times P^{0.16}}{h} \tag{11}$$

   where $E$ and $h$ are the coagulation kernel and the cloud depth, respectively, and $E$ is usually assumed to be a constant
(Wang et al., 2001). In this module, the washout is assumed to be related to precipitation intensity, the aerosol species and so on.



To briefly describe these methods, Table 1 lists the formulas. The theoretical estimated scavenging coefficients are labeled Theory. The field observations estimated by Eq (3) and (9) are labeled M1 and M2, respectively. The updated estimated method by Eq (10) is labeled as M2'. The modeling results are labeled M3, and these results are compared with different methods in section 3.

## 3 Results and Discussion

### 3.1 Impacts of below-cloud wet scavenging on aerosols

In this case, the total precipitation amount was relatively low, but the precipitation duration was long. SNA represented the majority of the ions in the rainwater, accounting for 73% of the total and their temporal variations are shown in Fig. 1. The precipitation duration is marked with the blue frame. In the early stage, marked with light blue stripes, the precipitation duration was long and the precipitation intensity was weak. In the later period, from 16:29 PM on November 20th to 1:18 AM on November 21st, the precipitation began to strengthen and is marked with the blue shading. Before this event, a severe haze occurred which exceeded the National Ambient Air Quality Standard (NAQQS, 75 µg.m$^{-3}$) (Shi et al., 2018). When rain occurred, both the aerosols in the air and the SNA concentration in the rainwater gradually decreased, especially during the later stage. It's clearly visible in Fig. 1 that all aerosol concentrations on the rainy day were much lower than the hourly average aerosol concentrations during the APHH campaign. Following the rain, SNA reached relatively stable and low values. For example, $SO_4^{2-}$ decreased from 70.6 to 25.2 mg. L$^{-1}$ (or a reduction of 64.3%) in the rainwater. Accordingly, aerosol sulfate and decreased by nearly 6 µg.m$^{-3}$ in the air.

The time series and averaged spectrum distribution of particle number size distributions measured by POPC, SPAMS and SMPS are shown in Fig. 2. With the help of three instruments, the size distributions cover a rather wide range, from 0.014 to 10.35 µm. The spectrum distribution exhibited unimodal distributions peaked in the size range of 20-90 nm. The spectrum distribution for SPAMS of $NO_3^-$ and $SO_4^{2-}$ both showed particularly high consistency in terms of variation patterns, magnitude and particle size distribution (Lang et al., 2016). And for POPC, the trend was also in consistent well with the coarse size of SPAMS. As shown from Fig. 2(a), for POPC and SMPS, the number concentration did not immediately decrease due to relatively weak precipitation intensity before 16:29 PM on November 20th. And in the later period, the number concentration decreased sharply and remained at a low level. It agreed well with the radar echo and precipitation intensity during this rain event. In order to investigate the BWSC, 16:29 PM on November 20th is taken as the before the rain occurs time in calculating the M1 and it will not repeat in following sections.

### 3.2 Multimethod comparison of BWSCs

For further analysis, the estimated BWSCs based on multiple methods were compared and shown in Fig. 3. As for the observational methods, e.g., M1, M2 and M2', there is no significant difference in the range of magnitude between them. More specifically, the observed BWSCs by original M2 are larger than the updated M2' method. However, M2' (5.7×10$^{-5}$, 8.9×10$^{-5}$



and $5.4\times10^{-5}$ s$^{-1}$ for NO$_3^-$, SO$_4^{2-}$ and NH$_4^+$) is much closer to the results of M1 (~$10^{-5}$ s$^{-1}$ for particle size in the range of 0.014-10.35 μm). Since M1 is based on the variations of the aerosol numbers below the cloud, it may be more suitable for the estimation of the BWSCs. It also indicates that the updated M2' is much more reasonable than the original M2 for estimation of BWSCs of various chemical species. In contrast, the theory calculated BWSC as $1.9\times10^{-6}$ s$^{-1}$ has an order of magnitude lower than the observational results. Considering the effects of thermophoresis, diffusiophoresis and electric charges, there is a wider range of three orders of magnitude ($10^{-6}$-$10^{-4}$ s$^{-1}$) (Wang et al., 2010;Wang et al., 2014c). In addition, the BWSC for M3 ($3.2\times10^{-6}$ s$^{-1}$) is also one order of magnitude lower than the field measurements. The low BWSC in CTMs can explain the underestimation of simulated wet deposition, which is mainly thought caused by chemical process, modeled precipitation and emission in previous studies (Wang et al., 2008;Ge et al., 2011). Thus, the observed M1 and M2' may revise the theory and M3 results in the future.

To further compare the BWSCs based on the particle size, the results in this study are compared with those of previous studies in Fig. 4. The size-resolved BWSCs of 0.014-0.74, 0.2-2.5 and 0.7-10.35 μm are the total number concentration by SMPS, SPAMS and POPC, respectively, which are within a certain range ($1.81\times10^{-5}$-$8.53\times10^{-5}$ s$^{-1}$). At approximately 0.2 μm (the lower limit detection of the multicomponent analysis), the M1 of 0.014-0.74 and 0.2-2.5 μm results have a gap that mainly originates from the use of different experimental instruments and their detection limits. However, the estimated results for larger sizes ($d_p > 3$ μm) by POPC have great fluctuation, mainly due to less number concentrations (< 2 cm$^{-3}$) and were considered as unreliable in this work. The BWSC from M1 showed a slowly decreasing trend in 0.014-0.2 μm and a significant increasing trend as $d_p > 0.2$ μm in this study, which is similar with the results of Huang Mountain (Wang et al., 2014c) and southern Finland (Laakso et al., 2003). Besides, the BWSC from M2' for SNA are similar to the results of M1 in 2.5 μm. Although there is a different trend with that reported in Lanzhou (Zhao et al., 2015) before 0.6 μm, both study exhibited an increasing trend after 0.6 μm. The difference of BWSCs from M1 in each sites may due to the measuring conditions (Wang et al., 2010). However, compared to the theory methods, this difference is very small as shown in Fig. 5. Different from the observational results, the theoretical results show a strong dependence on the particle size with obvious decreasing trend ($d_p < 1$ μm) and quickly increasing trend ($d_p > 1$ μm). As Seinfeld and Pandis (1998) mentioned, Brownian diffusion and gravity are the principal mechanisms affecting collection efficiency with $d_p$ smaller than 0.2 μm and larger than 2.5 μm, respectively. Theoretical estimation can effectively characterize the observed BWSC of aerosols in these two ranges. For the "Greenfield gap", there is large difference between the BWSC from M1 and theory method with the later is one order of magnitude lower. One reason is that all the influencing mechanisms still have not been fully considered and understood (Seinfeld, 1998), another reason is the existing ideal assumptions in derivation, such as no chemical reactions or emissions in the scavenging process; Ignored irregular surface of the aerosols and hygroscopic growth will increase the concentration of particles and then influence the scavenging efficiency (Wang et al., 2014c). Other extensive explanation is that the turbulent flow fluctuation, evaporation and breakup of raindrops are also important but neglected processes (Wang et al., 2010).





### 3.3 The parameterization of BWSCs

To discuss the uncertainties of the BWSC underestimation by theoretical calculations in different rainfall events, the data in summer of 2014 have also been included. As shown in Fig. 5, a strong relationship between the BWSCs and precipitation intensity obeys exponential power distribution both in summer of 2014 and the rainfall event in winter of APHH campaign in Beijing with the correlation coefficients for SNA are over than 0.68. Since the estimated BWSCs for SNA based on M1 and M2' in this event are in line with previous studies in summer, it indicated that the wet scavenging rule and regression fitting formulas are also universal in Beijing not only in summer but also in winter. In fact, this exponential power relationship has been confirmed in previous studies (Jylhä, 1991;Okita et al., 1996;Andronache, 2004a;Wang et al., 2014a;Wang et al., 2014b;Xu et al., 2017):

$$K = a \times P^b \tag{12}$$

where parameter $b$ represents the change rate of BWSCs along with $P$, while $a$ is equal to the WSCs when the $P = 1$ mm/h. Both $a$ and $b$ relate to chemical species and aerosols particle size.

For the further comparison, Fig.6 displays the parameterization of $a$ and $b$ in the exponential power relationship for BWSCs with the precipitation intensity by multimethod, i.e., theoretical method and field measurement methods. For the theory calculation, the parameter $a$ varies from a relatively wide range of $2.8 \times 10^{-8}$-$6.7 \times 10^{-5}$ s$^{-1}$ and BWSCs also have a wide range of 3-4 orders of magnitude for given precipitation intensities (Andronache, 2003;Wang et al., 2014b). Similar with the multimethod comparison of estimated BWSCs in this rainfall event of APHH campaign, parameterization for BWSCs obtained by M2/M2' shows higher magnitude of variations with the precipitation intensity with all of the straight line lie above the upper range of the theory calculation. It indicates recent theory calculated BWSCs has an obvious underestimation not only in a rainfall event but also in the parameterization of large number of rainfall events with different precipitation intensities and need revised or updated by the field measurement estimation.

### 3.4 Impacts and implications

To investigate the impacts of the wet scavenging on aerosol concentrations in the air and the wet depositions in rainfall, multimethod-estimated BWSCs included in Table 2 were used to rebuild the aerosol concentrations and wet depositions after one hour of rainfall event. Assuming the aerosol concentrations in the air are only influenced by the wet scavenging during rainfall event, its variation should be followed by Eq (13) according to Seinfeld and Pandis (1998):

$$\frac{dC_a}{dt} = -KC_a \tag{13}$$

$$C_a = C_{a0} e^{-Kt} \tag{14}$$

Variations of aerosol concentration can be resolved as Eq (14), in which the $K$ is a constant BWSC. Where, $K$ and $t$ are the BWSCs and the scavenging time, and $C_{a0}$ is the original aerosol concentration before the rainfall. In this study, the



$C_{a0}$   have been observed as 17.6 µg.m$^{-3}$, 9.8 µg.m$^{-3}$, 9.7 µg.m$^{-3}$ and 74.7 µg.m$^{-3}$ for NO$_3^-$, SO$_4^{2-}$, NH$_4^+$ and PM$_{2.5}$, respectively. After one hour of the wet scavenging by rainfall, the concentration of NO$_3^-$, SO$_4^{2-}$, NH$_4^+$ and PM$_{2.5}$ decreased to 14.3 µg.m$^{-3}$, 7.6 µg.m$^{-3}$, 7.9 µg.m$^{-3}$ and 65.2 µg.m$^{-3}$ respectively. Previous studies have confirmed that the exponential power distribution between the WSCs and precipitation intensity as Eq (12). And the size-resolved BWSC are accumulated for calculating the total BWSC for PM$_{2.5}$. As it is shown in Table 2, except CAMx, the calculated aerosol concentrations using theory and M3

BWSCs performed the obvious overestimation of PM$_{2.5}$ concentrations with the bias from 2.3 µg.m$^{-3}$ to 9 µg.m$^{-3}$, while showed similar with the observation for M1 and M2' BWSCs (bias < 1 µg.m$^{-3}$). It should be noted that the magnitude of BWSCs in the range of 10$^{-5}$-10$^{-4}$ perform the better-calculated aerosol concentrations than that the lower range. Wet deposition has also been reconstructed according to Eq (7), with the precipitation intensity setting as 0.17 mm/h, and the column height considering as 3 km. The Net Mean Bias (NMB) for the below-cloud wet depositions of NO$_3^-$ and NH$_4^+$ are -28% and -33%, while for SO$_4^{2-}$

is -49% according to the BWSCs in this study shown in Table 2.

     Overall, the M1 and updated M2' field observation results can effectively characterize the below-cloud scavenging ability whereas theory and M3 have obvious deviation. Therefore, the field measurements are needed to compensate for the defects in the theoretical and modeling calculations that provides room to make further progress in wet deposition numerical simulation.

## 4 Conclusions

An evaluation of below-cloud wet scavenging ability is first conducted based on field measurements, and accompanied with the theoretical estimation and modeling calculation. The averaged BWSCs obtained by field measurements are similar to each other of 10$^{-5}$ s$^{-1}$ and there exists strong exponential power relationship between BWSCs and precipitation intensity. Theoretical estimations coincide well with the observed BWSCs of aerosols with the   $d_p$   in ranges of smaller than 0.2 µm and larger than 2.5 µm, but are one magnitude lower than observed BWSCs within 0.2-2.5 µm. In the form of exponential power distribution

of BWSCs with precipitation intensity, the upper range of theoretical results is also lower than the measurement estimation. Thus, the underestimation of BWSC through theoretical method has been confirmed not only in APHH-Beijing campaign but also in all rainfall events in summer of 2014. These theoretical values are usually applied in CTMs with simplified scheme and accordingly the model calculations show lower BWSCs. It may explain the underprediction of the wet deposition both in global and in regional models of polluted regions. Field measurements are currently required to compensate for the theoretical

and modeling calculations and to construct a more reasonable and suitable simulation scheme to improve the wet deposition simulation, especially in polluted regions.

## Competing interests

The authors declare that they have no conflict of interest.



## Author contribution

DX, BG and ZW designed the whole structure of this work, XC performed the modeling calculation, YS, NC, ML, XP, ZM and YP prepared the SMPS data, the POPC data and the SPAMS data, respectively. DX performed the sequential sampling of rainwater, analyzed the data. DX and BG prepared the manuscript with contributions from all-authors.

## Acknowledgment

This study was financially supported by the National Natural Science Foundation of China (Grant Nos. 41575123, 41571130024, 41877313, and 41330422) and the National Key Research and Development Program of China (Grants 2017YFC0210103).



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




**Table 1. List of multimethod calculations for the BWSCs**

| | Formula | Reference | Symbol |
|---|---|---|---|
| Theory | $K(d_p) = \int_0^\infty \frac{\pi}{4} D_p^2 U_t(D_p) E(D_p, d_p) N(D_p) dD_p$ | Seinfeld and Pandis (1998), Wang et al. (2014c) | Theory |
| Field Observation | $K(d_p) = \frac{1}{t_1 - t_0} \ln[\frac{N_0(d_p)}{N_1(d_p)}]$ | Laakso et al. (2003) | M1 |
| | $K = \frac{C_p}{C_a(0) \times f} \times \frac{P}{h}$ | Andronache (2004b) | M2 |
| | $K = \frac{C_{p,below}}{C_a'(0) \times f} \times \frac{P}{h}$ | Xu et al. (2017) | M2' |
| Modeling calculation | $K = \frac{10.8 \times E \times P^{0.16}}{h}$ | Wang et al. (2001) | M3 |





**Table 2. The observed aerosol concentration and wet deposition, and parameters of the exponential power fittings, WSCs, rebuilt aerosol concentrations and wet depositions after one hour compare with multimethod.**

| Source | Species | $K = aP^b$ | | BWSC ($s^{-1}$) | Aerosol concentration ($\mu g \cdot m^{-3}$) | Wet deposition ($mg \cdot m^{-2}$) | Supplementary information |
|---|---|---|---|---|---|---|---|
| | | *a* | *b* | | | | |
| | $NO_3^-$ | | | | 14.3 | 3.2 | |
| | $SO_4^{2-}$ | | | | 7.6 | 5.9 | Observed aerosol concentrations and wet depositions after one-hour rainfall |
| | $NH_4^+$ | | | | 7.9 | 1.8 | |
| | $PM_{2.5}$ | | | | 65.2 | - | |
| | $NO_3^-$ | $2.5 \times 10^{-4}$ | 0.61 | $5.7 \times 10^{-5}$ | 14.4 | 2.3 | |
| This study [a] | $SO_4^{2-}$ | $7.6 \times 10^{-5}$ | 0.80 | $8.9 \times 10^{-5}$ | 7.1 | 3.0 | Winter in Beijing, M2' |
| | $NH_4^+$ | $1.1 \times 10^{-4}$ | 0.52 | $5.4 \times 10^{-5}$ | 8.0 | 1.2 | |
| This study [a] | PM | - | - | $4.1 \times 10^{-5}$ | 64.6 | - | 0.014-10.35 μm, winter in Beijing, M1 |
| Okita et al. (1996) [a*] | $SO_4^{2-}$ | $1.38 \times 10^{-4}$ | 0.74 | $3.8 \times 10^{-5}$ | 8.6 | 0.6 | Sado, Japan, winter of 1992, M2 |
| Andronache (2004b) [a*] | $SO_4^{2-}$ | $4.0 \times 10^{-4}$ | 0.81 | $9.5 \times 10^{-4}$ | 7.0 | 3.3 | AIRMoN in the USA, M2 |
| Yamagata et al. (2009) [a*] | $SO_4^{2-}$ | - | - | $9.8 \times 10^{-6}$- $2.5 \times 10^{-4}$ | 4.0-9.5 | 0.04-15.8 | The Arctic, late spring of 2004, M2 |
| Laakso et al. (2003) [a] | PM | - | - | $9.1 \times 10^{-6}$ | 72.3 | - | 0.01-0.51 μm, southern Finland, 6 years of observations, M1 |
| Wang et al. (2014c) [a] | PM | $4.2 \times 10^{-5}$ | 0.16 | $3.1 \times 10^{-5}$ | 66.7 | - | 0.01-1 μm, Huang Mountain, 2001 summer, M1 |
| Zhao et al. (2015) [a] | PM | - | - | $3.2 \times 10^{-5}$ | 66.6 | - | 0.01-10 μm, Lanzhou, 2012.9-2013.8, M1 |
| This study [b] | $PM_{2.5}$ | | | $1.9 \times 10^{-6}$ | 74.2 | - | Theory |





| Wang et al. (2014b) [b] | PM$_{2.5}$ | $3.83\times10^{-7}$-$6.89\times10^{-4}$ | 0.64-0.91 | $10^{-8}$-$10^{-2}$ | - | - | |
| This study [c] | PM$_{2.5}$ | - | - | $3.2\times10^{-6}$ | 73.8 | - | NAQPMS, M3 |
| Luo et al. (2019) [c] | PM$_{2.5}$ | - | - | $2.8\times10^{-5}$ | 67.5 | - | GEOS-Chem |
| ENVIRON.Inc (2005) [c] | PM$_{2.5}$ | $4.2\times10^{-4}$ | 0.79 | $1.8\times10^{-4}$ | 51.3 | - | CAMx |

[a] field observation, [b] theory and [c] modeling calculation

[*] WSC and [**] the wet scavenging effects on PM$_{2.5}$



**Figure captions**

**Figure 1. Hourly average aerosol concentration during November 11th to December 11th (box) and the rainy period on November 20th to 21st (red line and hollow circles) for (a) NO$_3^-$, (b) SO$_4^{2-}$ and (c) NH$_4^+$. The rainwater concentrations are shown as follows: (a) NO$_3^-$, blue; (b) SO$_4^{2-}$, red; and (c) NH$_4^+$, orange (line and triangles). Time series of the NO$_3^-$ (blue), SO$_4^{2-}$ (red) and NH$_4^+$ (orange) concentrations in the rainfall (lines and triangles) and in the air (lines) (d).**

**Figure 2. Time series of particle number size distributions (a) are measured by POPC, SPAMS (take SO$_4^{2-}$ for example) and SMPS, respectively. The averaged spectrum distribution of number concentration during the APHH-Beijing campaign (a) for SMPS (purple line), POPC (green line) and NO$_3^-$ (blue line), SO$_4^{2-}$ (red line) and NH$_4^+$ (orange line) by SPAMS.**

**Figure 3. Multimethod estimation of the BWSCs.**

**Figure 4. Multimethod estimation of the BWSCs and comparisons with previous studies.**

**Figure 5. Scatter plots of the BWSCs and precipitation intensity for NO$_3^-$ (a), SO$_4^{2-}$ (b) and NH$_4^+$ (c) (black dots: M2' in summer by Xu et al. (2017), light blue triangle: M2' in this case, deep blue triangle: M1).**

**Figure 6. The parameterization of BWSCs with the rainfall intensities.**

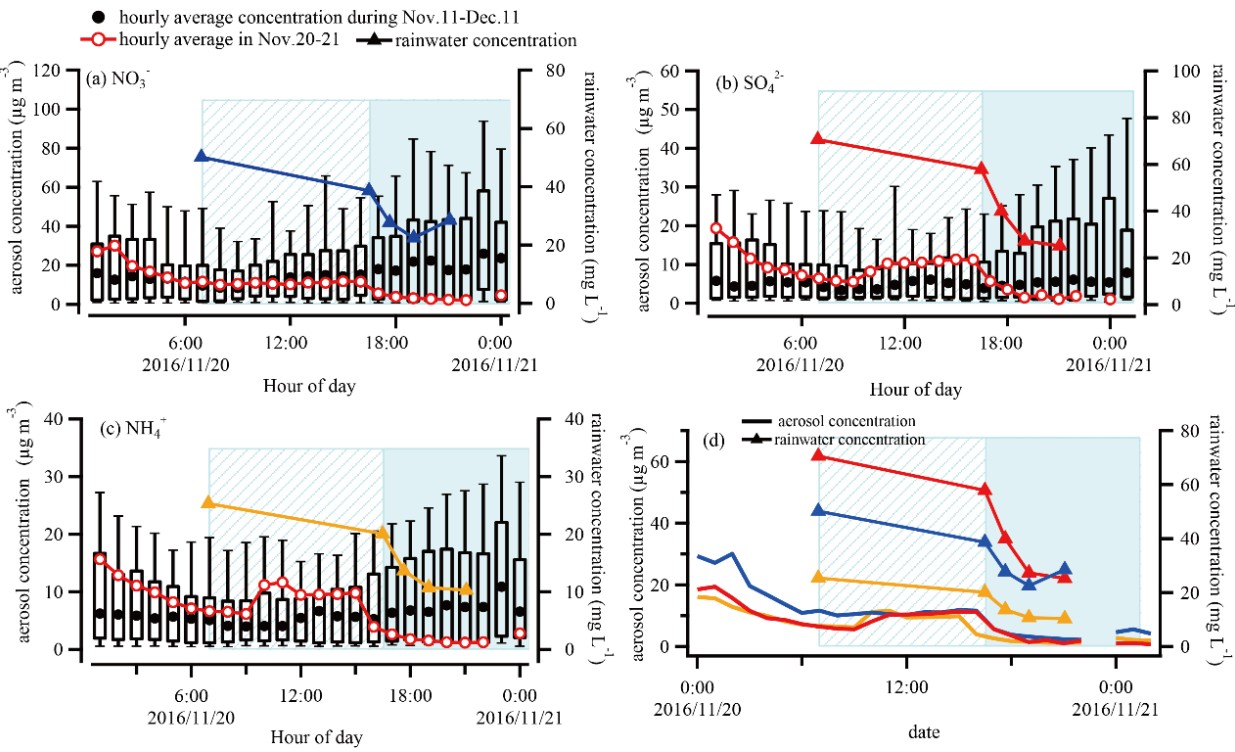

**Figure 1. Hourly average aerosol concentration during November 11th to December 11th (box) and the rainy period on November 20th to 21st (red line and hollow circles) for (a) NO$_3^-$, (b) SO$_4^{2-}$ and (c) NH$_4^+$. The rainwater concentrations are shown as follows: (a) NO$_3^-$, blue; (b) SO$_4^{2-}$, red; and (c) NH$_4^+$, orange (line and triangles). Time series of the NO$_3^-$ (blue), SO$_4^{2-}$ (red) and NH$_4^+$ (orange) concentrations in the rainfall (lines and triangles) and in the air (lines) (d).**





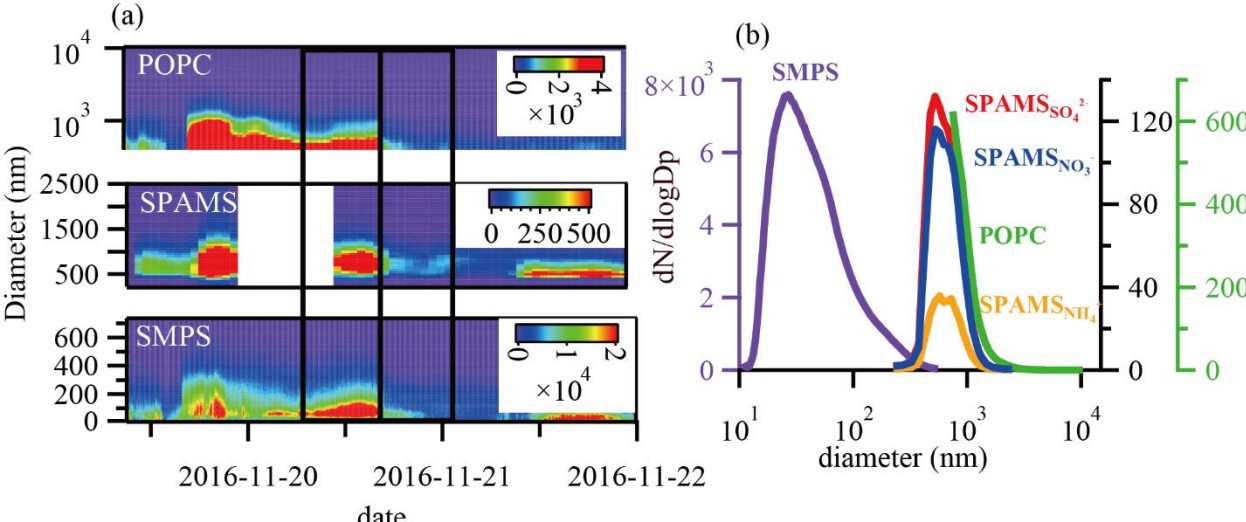

**Figure 2.** Time series of particle number size distributions (a) are measured by POPC, SPAMS (take $SO_4^{2-}$ for example) and SMPS, respectively. The averaged spectrum distribution of number concentration during the APHH-Beijing campaign (a) for SMPS (purple line), POPC (green line) and $NO_3^-$ (blue line), $SO_4^{2-}$ (red line) and $NH_4^+$ (orange line) by SPAMS.



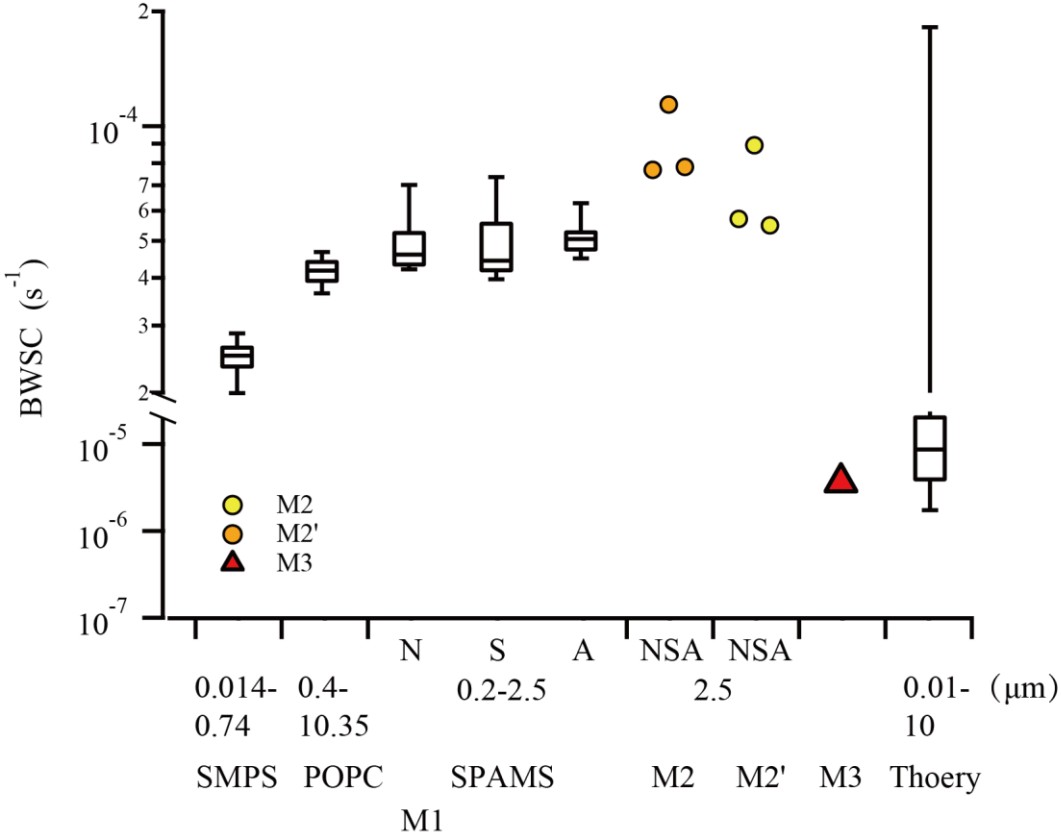

**Figure 3. Multimethod estimation of the BWSCs.**





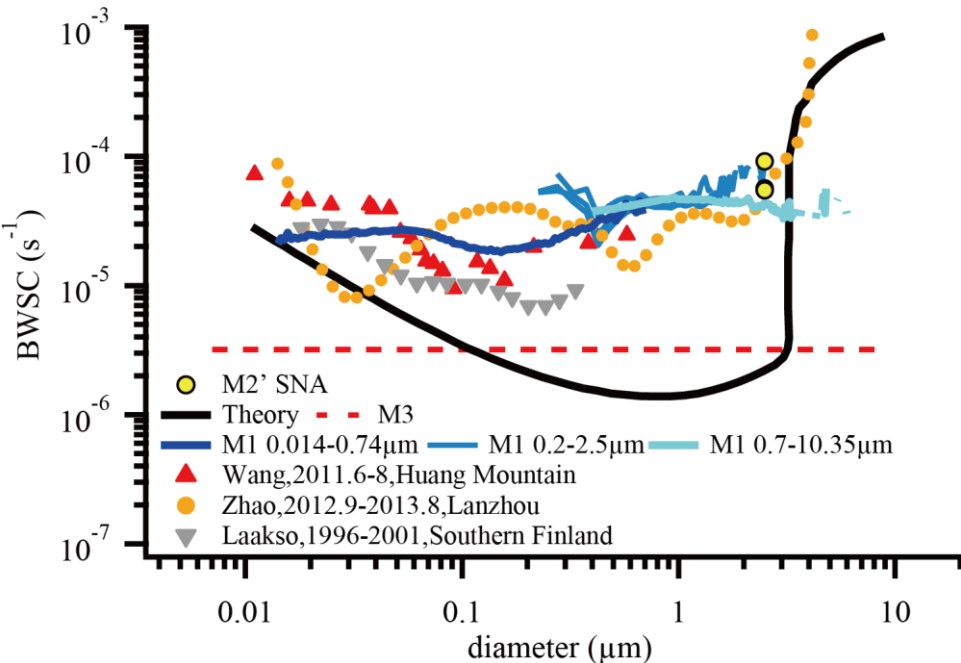

**Figure 4. Multimethod estimation of the BWSCs and comparisons with previous studies.**





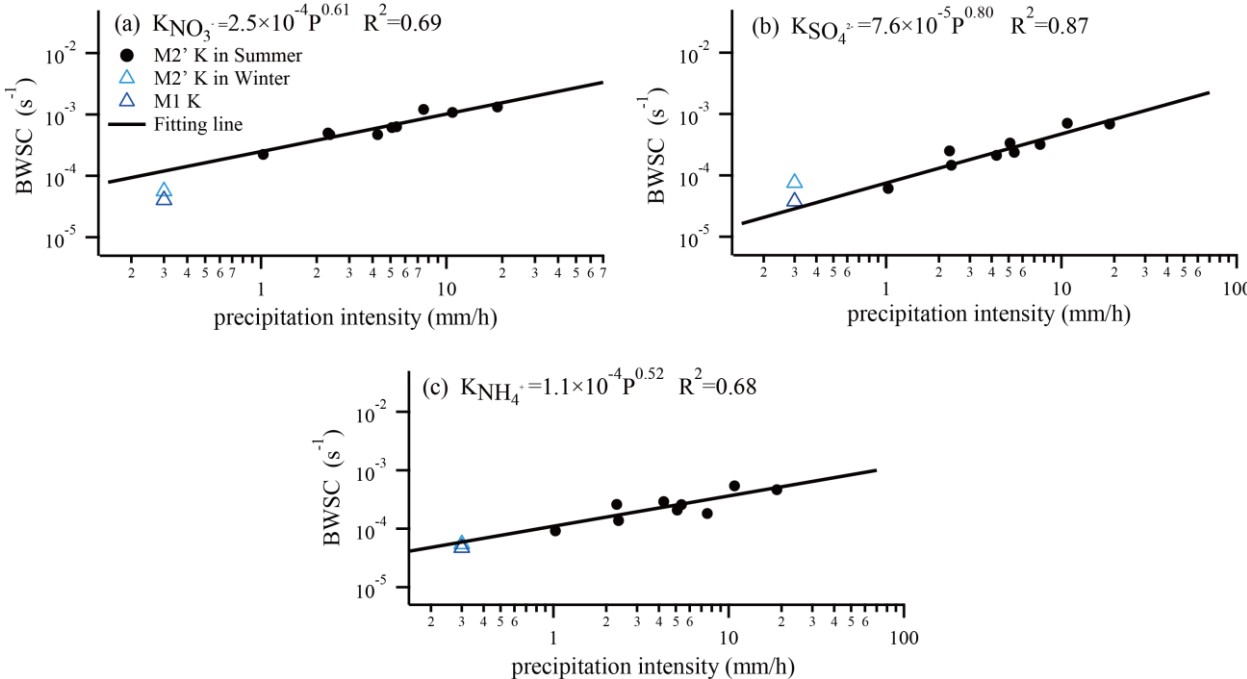

**Figure 5. Scatter plots of the BWSCs and precipitation intensity for NO₃⁻ (a), SO₄²⁻ (b) and NH₄⁺ (c) (black dots: M2' in summer by Xu et al. (2017), light blue triangle: M2' in this case, deep blue triangle: M1).**





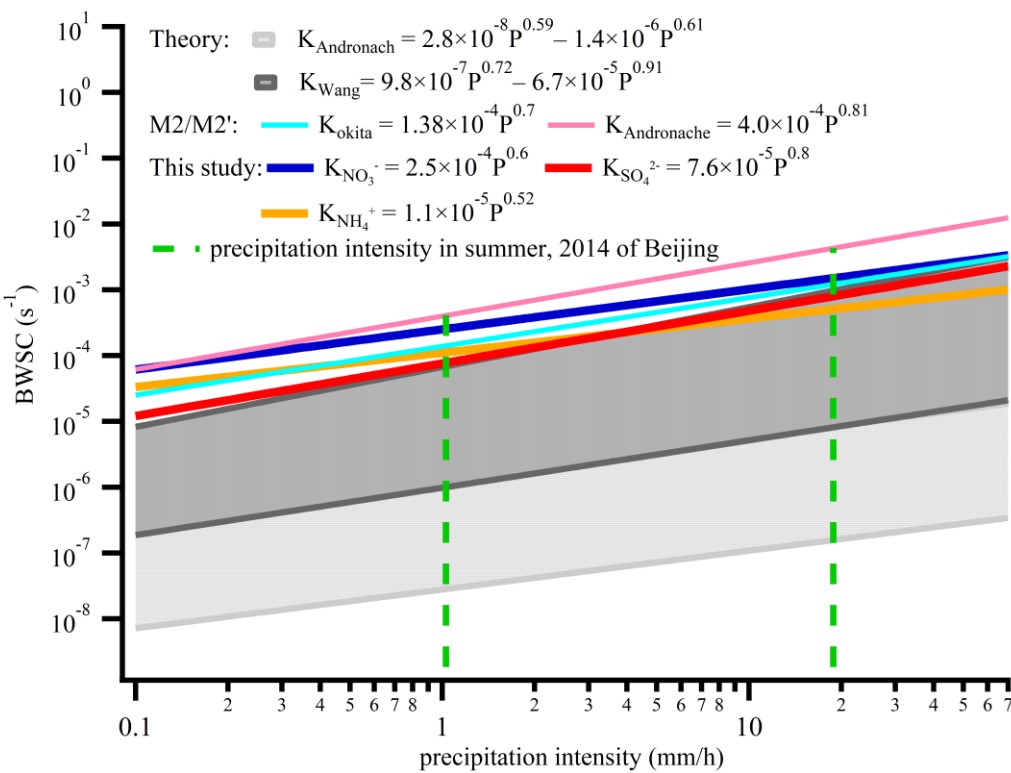

**Figure 6. The parameterization of BWSCs with the rainfall intensities.**
