# Peer review of "Multimethod determination of the below-cloud wet scavenging coefficients of aerosols in Beijing, China"

_Atmospheric Chemistry and Physics, 2019_

## Referee Comment (RC1) · Anonymous Referee #1 · 23 Aug 2019

This paper reports multimethod determination of the below-cloud wet scavenging coefficients of aerosols in Beijing, China. The analysis and interpretation of the results are overall fair. The paper presents very useful information regarding the wet deposition of aerosol. However, some additional information is still necessary for the readers to better understand this work.

Specific comments: 1. The modeling analysis presented in this study is subject to some uncertainties. For example, the washout was only parameterized for precipitation intensity, the aerosol species and so on. The coagulation kernel (E) was assumed to be a constant. What is the effect of this assumption on the modeling results? The authors

should estimate the uncertainties of their modeling analysis, or at least state clearly the model configuration and limitation so that the readers can judge by themselves. 2. A brief introduction to Fig.3 should be provided, which should be much helpful for the readers to better understand this study. 3. There are only two cases in the paper, more field measurement estimation could decrease the influence produced by accidental elements, and give more convincing results.
* * *

---

## Referee Comment (RC2) · Anonymous Referee #2 · 29 Aug 2019

The manuscript 'Multimethod determination of the below-cloud wet scavenging coefficients of aerosols in Beijing, China' written by Danhui Xu investigated wet scavenging process during APHH-Beijing. The authors tried to clarify well-known but not well-understood wet deposition process based on theoretical measure, field observation, and numerical model. The findings through this study is essential to understand the wet deposition process and also can contribute to refine the wet deposition scheme in CTMs and reduce its uncertainty. Although I would like to recommend to publish this manuscript, the following comments should be addressed with in-depth discussion for furthermore understanding.

**Major points:**

• P8, L192-193 (Symbols used in this manuscript):

I could follow the results and discussion section; however, it will be straightforward to use T, F (or O), and M to indicate theory, field observation, and modeling calculation, and these expressions are easy to catch discussion. Or, please include the rationale to use same M for field observation.

• P9, L220-227 (Discussion on BWSCs in Figure 3):

More discussion is required on the following two points.

What is M1 difference calculated by SMPS, POPC, and SPAMS? The particle size should be referred, though this can be partly covered in Figure 4. Please explicitly define the box plot shown in Figure 3. What percentiles (and/or average) are used to draw it?

M2 and M2' can generally agree with M1; however, the significant point should be the distinction of chemical species. From M1 of SPAMS, it seems no difference on SNA whereas M2 and M2' calculated larger BWSC for S. What are reasons to these results of SNA?

• P10, L256-257 (Relationship of BWSC and precipitation shown in Figure 5):

Data in 2014 summertime is essential to see this relationship; hence detail descriptions of this summer campaign is needed (will be appropriate introduced in Section 2.1). When compared winter results with summer results, it should be noticed that BWSC of NO3 (up to 10-3 sec-1) is greater than that of SO4 and NH4 during summertime. This is reversed finding during wintertime, because BWSC of SO4 is greater than NO3. What is reasons of these BWSC differences found in summer and winter?

Minor points:

- P2, L39: Do developed countries really have clean atmosphere?
- P2, L40-42: Which species are underestimated in MICS-Asia?
- P2, L42-44: How about sulfur species in TF-HTAP?
- P3, L47-48: Are these estimations suggested the importance of below-cloud scavenging based on what?
- P3, L49-51: Also, are these estimations based on what?
- P3, L47-51: If these estimations are based on model, how can we say its importance under the large uncertainty of modeling treatment?
- P3, L57: Does this sentence regarding gravity refer dry deposition?
- P4, L90-L92: I am not sure the meaning of "rarely in autumn winter" and "typical rainfall in winter", and these expressions will make some confusion. How rare the winter rain and what is the typical rain?
- P4, L111-L112: Here, "in the winter of 2016" indicates averaged VWA concentrations during APHH-Beijing 2016 campaign? It will be helpful to show the period (on L98). In addition, VWA concentrations during previous studies (Pan et al., 2013, 2012; Xu et al., 2017) can be added here for the discussion to strength the polluted episode analyzed in this study.
- P4, L118 and L122: Where is the sampling location of SPAMS and POPC? Same as wet-only sampler or SMPS?
- P4, L111-L123: I feel the summary of these observation dataset listed in one table (full name, abbreviation, target species, and short explanation, etc.) can help the reader to understand these observations used in this study.
- P5, L125-: The authors can put the notice that abbreviations are available in supplemental material.
- P7, L184-185: More information of NAQPMS will be needed. What kind of numerical model? Are there any previous studies show the modeling performance by NAQPMS? This can also motivate the authors to improve the model performance through this study.
- P8, L200-201 (Figure 1): Please include hourly (or finer time-scale) precipitation in Figure 1. We cannot catch the rainfall intensity from this sentence. Especially, we can see slight increase of aerosol concentration (except NO3) before 12:00 and what is the relation of this increase and precipitation? In my personal opinion, hourly concentration averaged during 11 November to 11 December is not needed here; remove this or add the discussion.

- P8, L208: From what to decreased by nearly 6 µg m-3 in the air? Need to add the variation about N and A.
- P9, L244: Typo of "Fig. 4"?
- P10, L259: Figure 5 shows R2, hence "coefficient of determination" is appropriate (or, Figure 5 include typo of R?).
- P10, L277-P11, L295 and caption of Table 2: To include the information of ambient concentration before the rainfall events (stated in P11, L285) is useful if explicitly stated in the caption.
- P11, L294: I guess that NMB is usually 'normalized mean bias'. Is this typo or some kind of another metric?

Technical comments:

- P8, L203: Typo of "NAAQS"?
- P13, L331-332: Check this reference style. There is no information related to volume, page, and year.
- P14, L354-356: Check this reference style regarding the location of year.
- P15, L390-391: This is not the latest edition.
- P16, L445-447: Check this reference style regarding the location of year.
- Throughout the manuscript, please unify "APHH" or "APHH-Beijing".
- Throughout the manuscript, please remove period "mg.m-3" and "µg.m-3".

---

## Author Comment (AC1) · 7 Nov 2019

The authors appreciate the reviewers for reviewing our manuscript and providing constructive comments. As suggested, we carefully revised the manuscript thoroughly according to the valuable advices, as well as the typographical, grammatical, and bibliographical errors. Listed below are our point-by-point responses in blue to the review's comments (in italic). The figures added in the reply is represented by 'Figure', which is distinguished from 'Fig.' in the manuscript.

**Anonymous Referee #1**

*This paper reports multimethod determination of the below-cloud wet scavenging coefficients of aerosols in Beijing, China. The analysis and interpretation of the results are overall fair. The paper presents very useful information regarding the wet deposition of aerosol. However, some additional information is still necessary for the readers to better understand this work.*

**[Response]:** Thanks for your suggestions and we have added the necessary information marked in the blue in the manuscript, such as section 2.2.3 (the description of modeling calculation), section 3.1 (the detailed introduction of this rain event) and section 3.2 (the detailed introduction of Fig. 3).

**Specific comments:**

*1.The modeling analysis presented in this study is subject to some uncertainties. For example, the washout was only parameterized for precipitation intensity, the aerosol species and so on. The coagulation kernel (E) was assumed to be a constant. What is the effect of this assumption on the modeling results? The authors should estimate the uncertainties of their modeling analysis, or at least state clearly the model configuration and limitation so that the readers can judge by themselves.*

**[Response]:** In the NAQPMS model, the below-cloud scavenging module from Comprehensive Air Quality Model with Extensions (CAMx) v4.42 was employed to calculate the below-cloud wet scavenging process and the wet scavenging coefficient was briefly described as follows:

$$K = \frac{4.2 \times 10^{-7} \times E \times P}{d_p} \tag{11}$$

where $d_p$ is the mean rain drop size and related to precipitation intensity. The collision efficiency $E$ is a function of aerosol particle size and mainly considers Brownian diffusion, interception and inertial impaction as shown in Figure. 1. According to Figure 1, E depends on the particle size, i.e. decreasing sharply as the diameter smaller than 400 nm, and slightly increasing from $10^{-4}$ to $10^{-3}$ when the particle size between 400 nm-2 μm, after that it increasing very quickly. Thus, choose the $E$ for different particle size will cause uncertainties in BWSC of one to two orders of magnitude for particle size in range of 0.01-2.5 μm. NAQPMS used in this study assumed SNA resides in fine mode size range (0.1-2.5 μm) and the geometric mean diameter of 0.5 μm was used in the calculation of $E$.

[Figure]

Figure 1. Size-resolved coagulation kernel based on the Brownian diffusion, interception and inertial impaction

We revised the model configure as "*The below-cloud scavenging module from Comprehensive Air Quality Model with Extensions (CAMx) v4.42 was employed to calculate the below-cloud wet scavenging process and the wet scavenging coefficient was briefly described as follows (Environ, 2005):*

$$K = \frac{4.2 \times 10^{-7} \times E \times P}{d_p} \tag{11}$$

*where $d_p$ is the mean rain drop size and related to precipitation intensity. The collision efficiency $E$ is a function of aerosol particle size and mainly considers Brownian diffusion, interception and inertial impaction. NAQPMS used in this study assumed SNA resides in fine mode size range (0.1-2.5 µm) and the geometric mean diameter of 0.5 µm was used in the calculation of $E$."*

And we also added the more description of NAQPMS in section 2.2.3 as "*In this study, a three-dimensional regional model, the Nested Air Quality Prediction Modeling System (NAQPMS) was adopted to calculate the aerosol scavenging coefficient. The NAQPMS, developed by IAP, is a fully modularized chemical transport model describing regional and urban-scale air pollution (Wang et al., 2001). The meteorological condition is driven by Weather Research and Forecasting (WRF) model. The NAQPMS consists of modules used for horizontal and vertical advection (Walcek and Aleksic, 1998), diffusion (Byun and Dennis, 1995), dry and wet deposition (Zhang et al., 2003; Stockwell et al., 1990), gaseous phase, aqueous phase, and heterogeneous atmospheric chemical reactions (Zaveri and Peters, 1999; Stockwell et al., 1990; Li et al., 2012). Carbon-Bond Mechanism Z (CBM-Z) and aerosol thermodynamic equilibrium partition model (ISORROPIAI1.7) have been used to calculated the gas and inorganic aerosol process. The cloud-process and aqueous chemistry module from Community Multi-scale Air Quality (CMAQ) modeling system v4.7 have been coupled in model by Ge et al. (2014). More details can be found in Li et al. (2016, 2017a). The NAQPMS has been widely used in prediction of acid rain, dust and secondary pollutions and can also reproduce well the physical and chemical evolution of reactive pollutants by solving the mass balance equations in terrain-following coordinates (Chen et al.,2019; Yang et al., 2019). It has been applied in Ministry of Ecology and Environment and local Environmental Protection Bureau such as Beijing, Shanghai, Guangzhou and Nanjing, etc. The NAQPMS also made great contribution to air quality assurance during the major activities (Wang et al., 2001; Wang et al., 2014d; Wu et al., 2010).*"

*2. A brief introduction to Fig.3 should be provided, which should be much helpful for*

*the readers to better understand this study.*

**[Response]:** We added more introductions of the symbols in the caption to Fig. 3 as *"The top and bottom of the boxes represent the 75th and 25th percentiles, and central lines mean the median BWSCs. The whiskers represent maximum and minimum BWSCs, respectively."*

Besides, for better understanding of multimethod, we have revised the symbols for multimethod and unified the expressions in the manuscript as: *"The theoretical estimated scavenging coefficients are labeled T. The field observations estimated by Eq (3) and (9) are labeled O1 and O2, respectively. The updated estimated method by Eq (10) is labeled as O2'. The modeling results are labeled M. And in following comments, we use the updated symbols instead. It will not be repeated later."*

We also added the detailed description for Fig. 3 in section 3.2 as *"The observed O1 by SMPS, which mostly covers the range of Aitken and accumulation mode aerosols (0.014-0.74 μm), are much lower than the other two measurements. The observed BWSCs by original O2 are larger than the updated O2' method. However, O2' ($5.7 \times 10^{-5}$, $8.9 \times 10^{-5}$ and $5.4 \times 10^{-5}$ $s^{-1}$ for $NO_3^-$, $SO_4^{2-}$ and $NH_4^+$) is much closer to the results of O1 (~$10^{-5}$ $s^{-1}$ for particle size in the range of 0.014-10.35 μm).*

*3. There are only two cases in the paper, more field measurement estimation could decrease the influence produced by accidental elements, and give more convincing results.*

**[Response]:** We agree with the reviewer's comment as well. In order to make up the bias from the limited rain events, we added the nine rain events at the same sampling site in IAP in summer of 2014 to compare the estimation methods of BWSCs of different precipitation with the same method O2'. Our results found that there is a strong exponential power relationship between the BWSCs and precipitation intensity both in the summer of 2014 and the rainfall event in winter of APHH-Beijing campaign with the coefficients of determination for SNA are over than 0.68. It indicated that rainfall event in winter of APHH-Beijing campaign also obeys the general wet scavenging rule in Beijing.

Although we cannot obtain the SMPS, SPAMS and POPC data in the summer of

2014 to compare the BWSCs under the multimethod, the same method the O2' has been employed to estimate the BWSCs in multi-event. Still, the multimethod to determinate the BWSCs of one rain event and even for multimethod of different rainfall events are needed for the future research.

**Reference:**

[revised manuscript text omitted]

---

## Author Comment (AC2) · 7 Nov 2019

The authors appreciate the reviewers for reviewing our manuscript and providing constructive comments. As suggested, we carefully revised the manuscript thoroughly according to the valuable advices, as well as the typographical, grammatical, and bibliographical errors. Listed below are our point-by-point responses in blue to the review's comments (in italic). The figures added in the reply is represented by 'Figure', which is distinguished from 'Fig.' in the manuscript.

**Anonymous Referee #2**

*The manuscript 'Multimethod determination of the below-cloud wet scavenging coefficients of aerosols in Beijing, China' written by Danhui Xu investigated wet scavenging process during APHH-Beijing. The authors tried to clarify well-known but not well-understood wet deposition process based on theoretical measure, field observation, and numerical model. The findings through this study is essential to understand the wet deposition process and also can contribute to refine the wet deposition scheme in CTMs and reduce its uncertainty. Although I would like to recommend to publish this manuscript, the following comments should be addressed with in-depth discussion for furthermore understanding.*

**[Response]:** We thank the reviewer for giving the positive comments. With response to the comments, the revised manuscript with in-depth discussion is much easier to understand for the readers, especially in section 2.2.3 (the description of modeling calculation), section 3.1 (the detailed introduction of this rain event) and section 3.2 (the detailed introduction of Fig. 3). And we also added the observation data list, the introduction of the nine precipitations in summer of Beijing, 2014, and more abbreviations in the supplementary file.

**Major comments:**

*1. P8, L192-193 (Symbols used in this manuscript):*

*I could follow the results and discussion section; however, it will be straightforward to use T, F (or O), and M to indicate theory, field observation, and modeling calculation, and these expressions are easy to catch discussion. Or, please include the rationale to use same M for field observation.*

**[Response]:** For better understanding of multimethod, we have revised the symbols for multimethod and unified the expressions in the manuscript as: "*The theoretical estimated scavenging coefficients are labeled T. The field observations estimated by Eq (3) and (9) are labeled O1 and O2, respectively. The updated estimated method by Eq (10) is labeled as O2'. The modeling results are labeled M.*" And in following comments, we use the updated symbols instead. It will not be repeated later.

*2. P9, L220-227 (Discussion on BWSCs in Figure 3):*

*More discussion is required on the following two points.*

*- What is M1 difference calculated by SMPS, POPC, and SPAMS? The particle size should be referred, though this can be partly covered in Figure 4. Please explicitly define the box plot shown in Figure 3. What percentiles (and/or average) are used to draw it?*

**[Response]:** The M1, now is called O1, calculated by the number concentrations collected from SMPA, SPAMS and POPC, respectively. The SMPS mainly measure particle number concentration from 14 to 740 nm. The SPAMS mainly measure the various chemical compositions with particle diameters ranging from 0.2 to 2.5 μm, while the POPC mainly focus on the coarse diameter of 0.4-10.35 μm (Table. s1). For readers' convenience, we added the observation instruments message in Fig. 4 caption. And we also added the description as "*The observed O1 by SMPS, which cover the range of Aitken and accumulation mode aerosols (0.014-0.74 μm), are much lower than the other two measurements (0.2-2.5 μm for SPAMS and 0.4-10.35 μm for POPC, respectively).*"

In Fig. 3, we have added the detailed introduction as: "*Fig 3. Box and whisker plots of the multimethod estimation of the BWSCs. The top and bottom of the boxes represent the 75th and 25th percentiles, and central lines mean the median BWSCs. The whiskers represent maximum and minimum BWSCs, respectively.*"

*- M2 and M2' can generally agree with M1; however, the significant point should be the distinction of chemical species. From M1 of SPAMS, it seems no difference on SNA whereas M2 and M2' calculated larger BWSC for S. What are reasons to these results of SNA?*

**[Response]:** There are two main reasons. One is the selection of the time. As we mentioned in section 3.1, due to the light rain intensity in early stage (no more than 0.5 mm from 7:00 AM to 16:00 PM, hourly rainfall data have been added in the revised Figure 1), the data from 16:29 PM on November 20[th] is taken as the beginning time of the rainfall event in calculating the O1. The data from 7:00 AM to 16:00 PM was represented as "before rainfall events" and calculated the mean value for SNA as well as the aerosol numbers. However, the data from SPAMS, 3 hours data from 7:00 AM -10:00 AM were missing. The missing data may result some uncertainties in estimating the BWSC using O1. In general, the median BWSCs of $NO_3^-$, $SO_4^{2-}$ and $NH_4^+$ in different particles sizes are $4.6\times10^{-5}$, $4.4\times10^{-5}$ and $5.0\times10^{-5}$ $s^{-1}$ in O1 of SPAMS, respectively. However, for $PM_{2.5}$ (total aerosols with particle size smaller than 2.5 μm) of BWSCs estimated by O2' are $5.7\times10^{-5}$, $8.9\times10^{-5}$ and $5.4\times10^{-5}$ $s^{-1}$, respectively, which indicated larger BWSC for $SO_4^{2-}$. In spite of this, the two methods estimated similar BWSC for $NO_3^-$ and $NH_4^+$.

Another one is the different sampling site of SPAMS with the other analyzers (SMPS, POPC, etc). The SPAMS is deployed in the China National Environmental Monitoring Centre (CNEMC), which is located in the northeast, 8 km away from the IAP sampling site. This site is a typical suburban site and mainly affected by residential source. (Figure. 2). Before this rainfall event, Beijing was occurring an air pollution case. The measurement in IAP site found that the NOR/SOR (oxidation ratios of $NO_x$/oxidation ratios of $SO_2$) decreased to 0.84, while the $SO_4^{2-}/NO_3^-$ increased to 0.38, which implicated a special increasing of $SO_4^{2-}$ in this event. The especially high $SO_4^{2-}$ concentration is mainly due to speeding up of $SO_2$ oxidation inducing by the $NH_4NO_3$, which drives the increasing of aerosol water content and trigger the positive feedback between AWC and aerosol secondary aerosol formation (Ge et al., 2014). The high proportion of S in aerosols were much easier to be scavenged by the rainfall, as the data of $NO_3^-$, $SO_4^{2-}$ and $NH_4^+$ decreased from 50.1, 70.6 and 25.3 mg $L^{-1}$ to 28.5, 25.2 and 10.3 mg $L^{-1}$ (a reduction of 43.2, 64.3 and 59.5%) in the rainwater shown in Figure. s2. Therefore, the much higher BWSCs of $SO_4^{2-}$ estimated by O2' in IAP site was reasonable. We added the sampling site

information in section 2.1 as "*It's deployed during the measuring time in China National Environmental Monitoring Centre (CNEMC), which is located in the northeast, 8 km away from the IAP sampling site. This site is a typical suburban site and mainly affected by residential source.*"

[Figure]

Figure.2 The sampling site in IAP, Beijing (marked in red) and CNEMC (marked in blue)

*3. P10, L256-257 (Relationship of BWSC and precipitation shown in Figure 5):*

*Data in 2014 summertime is essential to see this relationship; hence detail descriptions of this summer campaign is needed (will be appropriate introduced in Section 2.1). When compared winter results with summer results, it should be noticed that BWSC of NO3 (up to $10^{-3}$ $sec^{-1}$) is greater than that of SO4 and NH4 during summertime. This is reversed finding during wintertime, because BWSC of SO4 is greater than NO3. What is reasons of these BWSC differences found in summer and winter?*

**[Response]:** We added the detail descriptions of precipitation in summer of 2014 in supplementary file as "*The Figure. s2 shows the average concentrations of SNA in summer of 2014 (Box and whisker plot) and this rain event in winter of APHH-Beijing campaign. The VWA concentrations are no more than 30, 40 and 15 mg $L^{-1}$ for $NO_3^-$, $SO_4^{2-}$ and $NH_4^+$, respectively, and decreased sharply during the beginning of rainfall and remained at low levels during the event. The in-cloud scavenging process is*

*considered as the median value of the concentrations after accumulated precipitation exceeds 5 mm. These values were 2.75, 3.33, and 2.51 mg L$^{-1}$ for NO$_3^-$, SO$_4^{2-}$ and NH$_4^+$, respectively, in summer of 2014 (as shown in Figure. s2 marked in grey shadow) (Xu et al., 2017).*"

In normal case, NO$_3^-$ bearing compounds in coarse particles are much more easily scavenged than SO$_4^{2-}$ and NH$_4^+$ (Xu et al., 2017) same with the fitting line as shown in previous results in summer of 2014 in Fig. 5. However, it is different in this rainfall event in winter of APHH-Beijing campaign. According to the calculation of O2', the larger difference of concentration between the beginning fraction and the later fraction in rainfall events, the more efficiently of the below-cloud scavenging will be estimated. In this rain event, NO$_3^-$, SO$_4^{2-}$ and NH$_4^+$ decreased from 50.1, 70.6 and 25.3 mg L$^{-1}$ to 28.5, 25.2 and 10.3 mg L$^{-1}$ (a reduction of 43.2, 64.3 and 59.5%) in the rainwater (Figure. s2), with the largest ratio of reduction for SO$_4^{2-}$. Before this rainfall event, Beijing was occurring an air pollution case. The measurement in IAP site found that high SO$_4^{2-}$ concentration in aerosols. For example, the NOR/SOR (oxidation ratios of NO$_x$/oxidation ratios of SO$_2$) decreased to 0.84, while the SO$_4^{2-}$/NO$_3^-$ increased to 0.38 from pre-polluted period to pollution period. The especially high SO$_4^{2-}$ concentration is mainly due to speeding up of SO$_2$ oxidation inducing by the NH$_4$NO$_3$, which drive the increasing of aerosol water content and trigger the positive feedback between AWC and aerosol secondary aerosol formation (Ge et al., 2014). The high proportion of S in aerosols were much more easier to be scavenged by the rainfall. Therefore, the much higher BWSCs of SO$_4^{2-}$ estimated by O2' in this event was reasonable.

However, due to the uncertainties of limited rainfall event during this campaign, the comparison of the multi-method estimation of BWSC needed more strong evidence as well as more field measurement data in various polluted conditions in the future.

[Figure]

Figure s2. Evolution of the (a) $NO_3^-$ (blue), (b) $SO_4^{2-}$ (red) and (c) $NH_4^+$ (orange) of precipitation during summer 2014 and November $20^{th}$ to $21^{st}$ within different precipitation fractions of several sampled precipitation events (The data show the lowest, lowest 25 percentiles, median highest quartile, highest 75 percentiles, and highest value, respectively).

**Minor points:**

*1. P2, L39: Do developed countries really have clean atmosphere?*

**[Response]:** No, sorry for our un-precise saying. And we revised this sentence as "*This may be true in most clean atmosphere, e.g., some clean regions where air pollutants in the boundary layer were not sufficient.*"

*2. P2, L40-42: Which species are underestimated in MICS-Asia?*

**[Response]:** "*Recently, some regional models in MICS-Asia (Model Inter-Comparison Study for Asia) obviously underestimated $SO_4^{2-}$ and $NO_3^-$ wet deposition in East Asia.*" And we also added the species message in the manuscript.

*3. P2, L42-44: How about sulfur species in TF-HTAP?*

**[Response]:** The previous studies point the overestimated modeled values of volume weighted averaged (VWA) sulfur (S) in Europe and North America while underestimated S concentration values in Asia (Vet et al., 2014). And we also added the messaged in this manuscript as follows: "*For global model assessment by*

*Hemispheric Transport of Atmospheric Pollutants (TF-HTAP), wet depositions of nitrogen were also underpredicted in region of North America, Europe and Asia where measured the high level of volume weighted averaged (VWA) nitrogen (N) concentrations in rainfall as > 1.25 mg N $L^{-1}$, as well as underestimated sulfur wet deposition in Asia (Vet et al., 2014).*"

*4. P3, L47-48: Are these estimations suggested the importance of below-cloud scavenging based on what?*

**[Response]:** According to the sequential sampling field measurements in polluted region, such as India and North China, below-cloud scavenging contributed to more than 53%. It indicated that below-cloud scavenging process is crucial. And in order to make the manuscript more accurate we revised as "*which contributed to more than 53% of the total wet deposition in some polluted areas such as India (Chatterjee et al., 2010) and North China (Ge et al., 2016; Xu et al., 2017) on the basis of sequential sampling field measurements.*"

*5. P3, L49-51: Also, are these estimations based on what?*

**[Response]:** These estimations are not only based on the field measurement but also on the modeling calculation. We also revised this sentence as "*can remove 50-80% of the number or mass concentration of below-cloud aerosols both by filed measurements and modeling calculation.*"

*6. P3, L47-51: If these estimations are based on model, how can we say its importance under the large uncertainty of modeling treatment?*

**[Response]:** The model calculation indeed has large uncertainty of one to two orders of magnitude for BWSC estimation (Wang et al., 2010; Wang et al., 2014b). However, not only the field measurement but also modeling calculation showed that precipitation could scavenge 50-80% of the number or mass concentration of below-cloud aerosols. Combining these two methods, it reveals the importance of the below-cloud wet scavenging.

*7. P3, L57: Does this sentence regarding gravity refer dry deposition?*

**[Response]:** Yes, but in this manuscript we mainly focus on the wet deposition, we

prefer to delete this sentence "*Especially coarse particles ( $d_p$ > 20 μm) are also easily scavenged through the effect of gravity*".

*8. P4, L90-L92: I am not sure the meaning of "rarely in autumn winter" and "typical rainfall in winter", and these expressions will make some confusion. How rare the winter rain and what is the typical rain?*

**[Response]:** The "rare" means the precipitation is less in autumn and winter than summer. A lot of literatures mentioned that more than 80% rain events in Beijing concentrate in summer (Chen et al., 2013; Han et al., 2019). The "typical" means the moving path of the precipitation as usually used in previous studies in Beijing. Previous studies have mentioned that precipitations usually generate in the northwest mountainous area of Beijing and move along the steering flow to southeast (Xiao et al., 2015). According to the radar base reflectivity in this case, the rain event gradually moved from northwest to southeast, which is same as the "typical" rainfall moving path. To avoid the ambiguity, we revised manuscript as follows: "*In North China, precipitations were mainly concentrated in summer (more than 80%) but rare in autumn and winter (Xu et al., 2008; Gao et al., 2015; Chen et al., 2013; Han et al., 2019). However, the air pollution events were usually occurred in autumn and winter in North China Plain (NCP). Here we select a typical rainfall event moving from northwest to southeast in winter of Beijing...*"

*9. P4, L111-L112: Here, "in the winter of 2016" indicates averaged VWA concentrations during APHH-Beijing 2016 campaign? It will be helpful to show the period (on L98). In addition, VWA concentrations during previous studies (Pan et al., 2013, 2012; Xu et al., 2017) can be added here for the discussion to strength the polluted episode analyzed in this study.*

**[Response]:** No, here in line 111-112 means the winter of 2016. As we mentioned in last comment, the rain event in winter is rare and there's only one rain event during the APHH-Beijing campaign (The case we analyze in this study). In order to strengthen the comparison, we added previous studies' results as follows: "*much higher than the VWA concentrations in the winter of 2016 (8.3, 9.5 and 4.1 mg L$^{-1}$,*

*respectively) and in previous studies in Beijing (6.3, 9.1 and 4.9 mg $L^{-1}$ in Pan et al. (2012, 2013) and 6.2, 7.9 and 4.6 mg $L^{-1}$ in Xu et al.(2017) of summer)."*

*10. P4, L118 and L122: Where is the sampling location of SPAMS and POPC? Same as wet-only sampler or SMPS?*

Reply: The sampling site of POPC is same as the SMPS. However, the SPAMS is deployed in National Environmental Monitoring Centre (CNEMC), which is located in the northeast, 8 km away from the IAP sampling site. This site is a suburban site and mainly affected by residential source. For better reading, we also supplemented the message in the manuscript as "*A single-particle aerosol mass spectrometer (SPAMS) can accurately characterize aerosol particles containing various chemical compositions with diameters ranging from 0.2 to 2.5 μm. It's deployed during the measuring time in National Environmental Monitoring Centre (CNEMC), which is located in the northeast, 8 km away from the IAP sampling site. This site is a typical suburban site and mainly affected by residential source. More detailed fundamentals of the SPAMS and description can be found in Li et al. (2011), Lin et al. (2017) and Cheng et al. (2018). Size-resolved airborne $NO_3^-$, $SO_4^{2-}$ and $NH_4^+$ are the main focuses in this study, and the time resolution is 1 hour. In the meantime, a polarization optical particle counter (POPC) is also deployed to obtain coarse particle (0.4-10.35 μm) size distribution at the IAP sampling site, and time resolution is 5-min. Detailed description and settings can be found in Tian et al. (2018).*"

*11. P4, L111-L123: I feel the summary of these observation dataset listed in one table (full name, abbreviation, target species, and short explanation, etc.) can help the reader to understand these observations used in this study.*

**[Response]:** We added the summary of the observation dataset in the supplementary file as follows:

Table s1. The observation dataset list

| Observation instrument | Abbreviation | Introduction (time-resolution and mainly measured material) |
|---|---|---|
| Ion Chromatography | IC | anions ($SO_4^{2-}$, $NO_3^-$, $Cl^-$ and $F^-$) and cations ($NH_4^+$, $Na^+$, $K^+$, $Ca^{2+}$ and $Mg^{2+}$) in the |

| | | rainfall samples |
|---|---|---|
| Ambient Ion Monitor-Ion Chromatograph | AIM-IC | 60 min resolution, $PM_{2.5}$ concentrations |
| Scanning Mobility Particle Sizer | SMPS | 5 min resolution, 14-740 nm particle number concentration |
| Single-particle Aerosol Mass Spectrometer | SPAMS | 60 min resolution, various chemical compositions with 0.2-2.5 μm particle number concentration, mainly focus on $NO_3^-$, $SO_4^{2-}$ and $NH_4^+$ |
| Polarization Optical Particle Counter | POPC | 5 min resolution, 0.4-10.35 μm particle number concentration |

*12. P5, L125-: The authors can put the notice that abbreviations are available in supplemental material.*

**[Response]:** The abbreviations have added in the supplementary file.

*13. P7, L184-185: More information of NAQPMS will be needed. What kind of numerical model? Are there any previous studies show the modeling performance by NAQPMS? This can also motivate the authors to improve the model performance through this study.*

**[Response]:** We have added more description of the NAQPMS: "*In this study, a three-dimensional regional model, the Nested Air Quality Prediction Modeling System (NAQPMS) was adopted to calculate the aerosol scavenging coefficient. The NAQPMS, developed by IAP, is a fully modularized chemical transport model describing regional and urban-scale air pollution (Wang et al., 2001). The meteorological condition is driven by Weather Research and Forecasting (WRF) model. The NAQPMS consists of modules used for horizontal and vertical advection (Walcek and Aleksic, 1998), diffusion (Byun and Dennis, 1995), dry and wet deposition (Zhang et al., 2003; Stockwell et al., 1990), gaseous phase, aqueous phase, and heterogeneous atmospheric chemical reactions (Zaveri and Peters, 1999; Stockwell et al., 1990; Li et al., 2012). Carbon-Bond Mechanism Z (CBM-Z) and aerosol thermodynamic equilibrium partition model (ISORROPIAI1.7) have been used to calculated the gas and inorganic aerosol process. The cloud-process and aqueous chemistry module from Community Multi-scale Air Quality (CMAQ)*

*modeling system v4.7 have been coupled in model by Ge et al. (2014). More details can be found in Li et al. (2016, 2017a). The NAQPMS has been widely used in prediction of acid rain, dust and secondary pollutions and can also reproduce well the physical and chemical evolution of reactive pollutants by solving the mass balance equations in terrain-following coordinates (Chen et al.,2019; Yang et al., 2019). It has been applied in Ministry of Ecology and Environment and local Environmental Protection Bureau such as Beijing, Shanghai, Guangzhou and Nanjing, etc. The NAQPMS also made great contribution to air quality assurance during the major activities (Wang et al., 2001; Wang et al., 2014d; Wu et al., 2010).*"

*14. P8, L200-201 (Figure 1): Please include hourly (or finer time-scale) precipitation in Figure 1. We cannot catch the rainfall intensity from this sentence. Especially, we can see slight increase of aerosol concentration (except NO3) before 12:00 and what is the relation of this increase and precipitation? In my personal opinion, hourly concentration averaged during 11 November to 11 December is not needed here; remove this or add the discussion.*

**[Response]:** We have added the hourly precipitation intensity and detailed captions in Fig. 1. Clearly can be seen that the precipitation is weak in the light rain period (no more than 0.5 mm from 7:00 AM to 16:00 PM) with the slight increase of aerosol concentration. For better comparison with variation of SNA in precipitation, hourly concentration of SNA in aerosols have been displayed in Figure 1. Also compared with the monthly data, the sharply decreasing after 16:00 PM indicated that the impacts of below-cloud scavenging on aerosols concentrations. Based on this intention, the hourly averaged concentration during the APHH-Beijing was remained. As suggested, the hourly precipitation intensity are added and the manuscript is revised as "*It's clearly visible in Fig. 1 that all aerosol concentrations on the rainy day were much lower than the hourly averaged aerosol concentrations during the APHH-Beijing campaign, especially during the precipitation time indicating the below-cloud scavenging impacts.*"

*15. P8, L208: From what to decreased by nearly 6 μg m$^{-3}$ in the air? Need to add the variation about N and A.*

**[Response]:** "$NO_3^-$, $SO_4^{2-}$ and $NH_4^+$ decreased from 50.1, 70.6 and 25.3 mg $L^{-1}$ to 28.5, 25.2 and 10.3 mg $L^{-1}$ (or a reduction of 43.2, 64.3 and 59.5%) in the rainwater. Accordingly, aerosol nitrate, sulfate and ammonium decreased from 13.8, 8.3 and 8.4 µg $m^{-3}$ to 1.2, 2.2 and 0.1 µg $m^{-3}$ in the air (decreased by more than 6 µg $m^{-3}$)." And we have added this description in the manuscript.

*16. P9, L244: Typo of "Fig. 4"?*

**[Response]:** Yes, we have revised it.

*17. P10, L259: Figure 5 shows $R^2$, hence "coefficient of determination" is appropriate (or, Figure 5 include typo of R?).*

**[Response]:** In L259, P10, the description changed to "coefficient of determination".

*18. P10, L277-P11, L295 and caption of Table 2: To include the information of ambient concentration before the rainfall events (stated in P11, L285) is useful if explicitly stated in the caption.*

**[Response]:** We have added the original aerosol concentrations before the rainfall into Table 2 as shown in the manuscripts.

*19. P11, L294: I guess that NMB is usually 'normalized mean bias'. Is this typo or some kind of another metric?*

**[Response]:** Yes, it's a typo and we have revised it as "Normalized Mean Bias".

**Technical comments:**

*1. P8, L203: Typo of "NAAQS"?*

**[Response]:** Yes, we have revised it.

*2. P13, L331-332: Check this reference style. There is no information related to volume, page, and year.*

**[Response]:** We have revised it in the manuscript. "*Andronache, C., Grönholm, T., Laakso, L., Phillips, V., and Venäläinen, A.: Scavenging of ultrafine particles by rainfall at a boreal site: observations and model estimations, Atmos. Chem. Phys., 6, 4739-4754, 2006.*"

*3. P14, L354-356: Check this reference style regarding the location of year.*

**[Response]:** We have revised the location of the year. "*Gao, X. D., Chen, X Y., Ding,*

*Z. W., Yang, W. Q. Investigation of the variation of atmospheric pollutants from chemical composition of precipitation along an urban-to-rural transect in Beijing. Acta Scientiae Circumstantiae (in Chinese), 35(12): 4033-404, 2015.*"

*4. P15, L390-391: This is not the latest edition.*

**[Response]:** We have revised the reference to latest edition as: "*Seinfeld, J. H., and S.N. Pandis.: Atmospheric chemistry and physics: from air pollution to climate change. John Wiley and Sons, Inc., NY., Atmospheric Chemistry and Physics, 2016*", and also check this reference throughout the manuscript.

*5. P16, L445-447: Check this reference style regarding the location of year.*

**[Response]:** We have revised the location of the year. "*Xu, J., Zhang, X. L., Xu, X. B., Ding, G. A., Yan. P., Yu, X. L., Chen, H. B., Zhou, H. G. Variations and source identification of chemical compositions in wet deposition at Shangdianzi background station. Acta Scientiae Circumstantiae (in Chinese), 28(5):1001 -1006, 2008.*"

*6. Throughout the manuscript, please unify "APHH" or "APHH-Beijing".*

**[Response]:** We have unified the description of "APHH-Beijing" through the manuscript.

*7. Throughout the manuscript, please remove period "mg.m$^{-3}$" and "µg.m$^{-3}$".*

**[Response]:** We have removed the period in "mg.m$^{-3}$", "µg.m$^{-3}$" and "mg.L$^{-1}$" throughout the manuscript.

**Reference:**

[revised manuscript text omitted]

---

## Author Comment (AC3) · 7 Nov 2019

The comment was uploaded in the form of a supplement:
https://www.atmos-chem-phys-discuss.net/acp-2019-680/acp-2019-680-AC3-supplement.zip